# Yeast Bax Inhibitor (Bxi1p/Ybh3p) Is Not Required for the Action of Bcl-2 Family Proteins on Cell Viability

**DOI:** 10.3390/ijms241512011

**Published:** 2023-07-27

**Authors:** Marek Mentel, Miroslava Illová, Veronika Krajčovičová, Gabriela Kroupová, Zuzana Mannová, Petra Chovančíková, Peter Polčic

**Affiliations:** Department of Biochemistry, Faculty of Natural Sciences, Comenius University in Bratislava, Mlynská Dolina CH1, Ilkovičova 6, 84215 Bratislava, Slovakia

**Keywords:** yeast, *Saccharomyces cerevisiae*, apoptosis, endoplasmic reticulum, Bcl-2 family, regulated cell death, unfolded protein response

## Abstract

Permeabilization of mitochondrial membrane by proteins of the BCL-2 family is a key decisive event in the induction of apoptosis in mammalian cells. Although yeast does not have homologs of the BCL-2 family, when these are expressed in yeast, they modulate the survival of cells in a way that corresponds to their activity in mammalian cells. The yeast gene, alternatively referred to as *BXI1* or *YBH3*, encodes for membrane protein in the endoplasmic reticulum that was, contradictorily, shown to either inhibit Bax or to be required for Bax activity. We have tested the effect of the deletion of this gene on the pro-apoptotic activity of Bax and Bak and the anti-apoptotic activity of Bcl-XL and Bcl-2, as well on survival after treatment with inducers of regulated cell death in yeast, hydrogen peroxide and acetic acid. While deletion resulted in increased sensitivity to acetic acid, it did not affect the sensitivity to hydrogen peroxide nor to BCL-2 family members. Thus, our results do not support any model in which the activity of BCL-2 family members is directly affected by *BXI1* but rather indicate that it may participate in modulating survival in response to some specific forms of stress.

## 1. Introduction

In the course of evolution, living organisms developed multiple mechanisms to increase survival under various adverse conditions. These involve several forms of programmed cell death, including apoptosis in animals, to maintain tissues or populations of cells, as well as processes such as autophagy, unfolded protein response, or oxidative stress response to maintain the functionality of individual cells.

Proteins of the BCL-2 protein family are involved in the regulation of apoptosis. They integrate multiple upstream death- or survival-promoting signals, to which they respond by permeabilization of the outer mitochondrial membrane, leading to the release of the cytochrome c from mitochondria into the cytosol and activation of caspases [1]. The BCL-2 family comprises proteins that are characterized by the presence of at least one of four sequence motifs (BH1-BH4, Bcl-2 homology) homologous to the founding member of the family: Bcl-2. The family consists of both pro-apoptotic proteins, which promote membrane permeabilization, and anti-apoptotic proteins, such as Bcl-2 and Bcl-XL, that support cell survival by preventing it. The function of individual members of the family correlates with the presence of BH motifs. While anti-apoptotic proteins contain all four BH motifs, all pro-apoptotic members lack BH4. They are further divided into two subclasses, multidomain pro-apoptotic proteins, e.g., Bax and Bak, which contain BH1–BH3 and are required to directly permeabilize the membrane and BH3-only proteins, which only contain BH3 and act as sensors that activate Bax and Bak [2,3,4].

Although regulated cell death in yeast does rely on different pathways than in mammalian cells [5,6] and the yeast genome does not contain homologs of BCL-2 family genes, when mammalian pro-apoptotic members of the family Bax and Bak are expressed in yeast, they induce cell killing [7]. Furthermore, the killing of yeast cells by Bax or Bak can be modulated by both anti-apoptotic BCL-2 family members and BH3-only proteins [8,9,10,11]. This, indeed, makes yeast an attractive model for the investigation of the regulation of mammalian apoptosis [12,13]. In one of the early yeast-based screenings of mammalian genes that suppress the ability of Bax to kill yeast cells, a membrane protein localized in endoplasmic reticulum, BI-1 (Bax inhibitor-1), was identified [14]. Unlike anti-apoptotic members of the Bcl-2 family, BI-1 does not inhibit Bax by direct interaction. Rather it was shown to be a Ca^2+^ channel that participates in the regulation of unfolded protein response by IRE1 [15] but also in response to calcium imbalance, reactive oxygen species accumulation, and metabolic dysregulation [16].

BI-1 became a founding member of the protein superfamily TMBIM (Transmembrane Bax inhibitor containing motif) that comprises homologous proteins, which share the common membrane topology [17] but differ in their subcellular localization (e.g., Golgi apparatus, endoplasmic reticulum, and mitochondria) and most of which have anti-apoptotic activity [18,19].

In yeast *Saccharomyces cerevisiae*, a homolog of mammalian BI-1 was identified [20] and named *BXI1* (Bax-inhibitor 1) [21]. It was shown to enhance yeast survival under conditions that induce increased levels of unfolded proteins in the endoplasmic reticulum, such as high temperature, β-mercaptoethanol or tunicamycin treatment, suggesting that it may play a role in unfolded protein response [21].

Interestingly, the same protein was in an independent study identified by in silico screening as a protein containing a motif similar to the BH3 domain of the mammalian BCL-2 family proteins. Named *YBH3* (Yeast BH3-only), it was shown to potentiate the killing of cells by Bax, as mutant cells lacking a functional copy of the gene survived better under conditions of Bax expression [22]. The effect of Ybh3 on Bax-mediated killing was shown to be dependent on the presence of mitochondrial phosphate carrier Mir1. Furthermore, Ybh3 was shown to participate in regulated cell death induced by acetic acid and hydrogen peroxide [22].

In this study, we tested the effect of the deletion of *BXI1* and *MIR1* on the pro-apoptotic activity of Bax and Bak and the anti-apoptotic activity of Bcl-XL and Bcl-2, as well on survival after treatment with inducers of regulated cell death in yeast, hydrogen peroxide and acetic acid. Our results do not support any model in which the activity of BCL-2 family members is directly affected by *BXI1*.

## 2. Results

To evaluate the effect of Bxi1p on cell death in yeast induced by ectopic expression of Bax and Bak, the recombinant genes enabling the galactose-inducible expression of HA-tagged version of murine Bax and Bak were integrated into the genome of wild type (CML282) and *∆bxi1* (CML282∆bxi1) strains. As expected, the growth of the CML282-GHBax strain on media containing 0.1% or more galactose is seriously compromised (Figure 1) as compared to the growth of the control strain (CML282) due to the expression of Bax. The same growth defect on galactose-containing plates was also observed in the growth of CML∆bxi1-GHBax. The sensitivity of both strains to the galactose-induced effects of Bax was identical regardless of the presence of the wild-type *BXI1* allele (Figure 1). The same results were also observed with the strains expressing Bak.

The effect of the expression of Bax and Bak on yeast cell growth can be suppressed by simultaneous expression of anti-apoptotic proteins Bcl-XL or Bcl-2 [7,23,24]. To see whether or not the function of these anti-apoptotic proteins can be affected by the absence of Bxi1p, we further transformed the wild type and *∆bxi1* strains containing integrated Bax or Bak genes with plasmids, enabling the tetracycline-inducible expression of HA-tagged murine Bcl-XL or Bcl-2. While all the strains containing Bax and Bak genes failed to grow on galactose-containing plates without tetracycline analog, doxycycline, strains containing plasmids with Bcl-XL restored growth when at least 0.1 mg/L doxycycline was present in cultivation media (Figure 2). Absence (*∆bxi1*) or presence of functional *BXI1* allele did not affect the growth of these strains.

To test the role of mitochondrial phosphate carrier isoform Mir1 in the cell death modulating activity of Bxi1 and BCL-2 family proteins, the strains with additional deletion of the *MIR1* gene were prepared (CML∆mir1, CML∆bxi1∆mir1). As in the experiment described above, the genes enabling the expression of Bax or Bak were integrated into their genomes (CML∆mir1-GHBax, CML∆bxi1∆mir1-GHBax and CML∆mir1-GHBak, CML∆bxi1∆mir1-GHBak, respectively) and resulting strains were transformed with plasmids, enabling the expression of Bcl-XL (pCM252-HABcl-XL) or with control plasmids (pCM252).

The deletion of *MIR1* alone did not affect either the growth of strains expressing pro-apoptotic Bax (Figure 3a) nor did it affect the growth of strain co-expressing Bax and anti-apoptotic protein Bcl-XL (Figure 3b). The same results were obtained with Bak and Bcl-2.

As it is shown on Figure 4, the deletion of *MIR1* gene also did not affect neither the growth inhibition by Bax, nor the growth restoration by Bcl-XL in *∆bxi1* strains. These data together indicate that the absence of *BXI1*, *MIR1* or the combined absence of both does not affect the activity of BCL-2 family proteins in yeast.

As it was suggested that the *BXI1* may play a role in regulated cell death in yeast induced by acetic acid or hydrogen peroxide [22], we have also tested the sensitivity of CML∆bxi1 to the treatment with these two compounds. While the sensitivity of the deletion strain to hydrogen peroxide was essentially the same as the sensitivity of the wild type (Figure 5a), the deletion strain showed increased sensitivity to acetic acid (Figure 5b).

## 3. Discussion

Budding yeast is a unicellular eukaryotic organism that has been traditionally used as a model to study processes typical for eukaryotic cells, including mammalian cells. While many pathways present in mammalian cells are well conserved and have obvious homologs in yeast, others may differ. A typical representative of the latter is programmed cell death. Unlike in mammalian apoptosis, the pathways of regulated cell death in yeast do not rely on the permeabilization of mitochondrial membrane by BCL-2 family, and the yeast genome also does not encode for homologs of BCL-2 family proteins. Surprisingly, the yeast genome, as also genomes of other eukaryotic organisms, including those who do not have apoptotic pathway dependent on the BCL-2 family, does encode for a homolog of mammalian BI-1. Moreover, a yeast homolog contains a sequence motif similar to the BH3 domain of BCL-2 family proteins. The study suggested that, as the only yeast BH3-only protein, it may be required for the pro-apoptotic activity of Bax and Bak when ectopically expressed in yeast cells [22]. As this observation appears to contradict the ability of BI-1 to inhibit Bax, we tested the response of wild-type yeast strain and deletion mutant lacking a functional copy of *BXI1* to survive the expression of pro-apoptotic and anti-apoptotic BCL-2 family proteins. The employed expression system enabled us to modulate the expression of these proteins by varying the concentration of inducers (galactose and doxycycline) in cultivation media [9]. In these experiments, the growth of the *∆bxi1* strain did not differ from the growth of the wild-type strain, even in the conditions when the lowest concentration of inducers required for the phenotypic effect was used (e.g., 0.1% galactose or 0.1 μg/mL doxycycline). This indicates that the absence of Bxi1p affects neither the ability of Bax or Bak to kill cells nor the ability of Bcl-XL or Bcl-2 to rescue cell growth. The same system has been previously shown to be suitable to detect the activity of mammalian BH3-only proteins [10,11]. Our data, therefore, strongly indicate that the motif identified in Bxi1p is not a functional BH3 domain.

At the molecular level, the function of the BH3 domain in mammalian BCL-2 family proteins is to facilitate the protein–protein interactions between family members by binding to the hydrophobic pocket in the partner protein [25,26,27]. This hydrophobic pocket is present on multi-domain family members (e.g., Bax, Bak, Bcl-XL, Bcl-2) and is also formed by BH domains [28]. Therefore, the conclusion that Bxi1p is not a yeast BH3-only protein would, indeed, also be consistent with the absence of BCl-2 family proteins in yeast.

The killing of yeast cells by Bax has been shown to depend on the integration of Bax into mitochondrial membranes [9]. This integration then leads to the formation of pores in the membranes, which correspond to pores formed by Bax in mammalian mitochondria [29]. Unlike in mammalian cells, the subsequent release of cytochrome *c* from mitochondrial intermembrane space into cytosol is not required for cell killing [30,31]. Presence of these pores itself is likely incompatible with yeast cells survival, e.g., due to its interference with mitochondrial biogenesis [32]. The same efficiency of the killing of yeast cells by Bax in the presence and the absence of *BXI1*, observed in this work, then suggests that the ability of Bax to integrate to mitochondrial membranes and to form pores is not affected by Bxi1p.

As we observed that the sensitivity of all tested deletion strains to the action of BCL-2 family proteins was the same as in the wild-type cells, our data not only indicate that *BXI1* is not required for the activity of BCL-2 family proteins in yeast but also imply that the absence Bxi1p or its presence in normal level, i.e., not overexpressed, do not affect the downstream responses of yeast cells that may affect the survival in this situation.

Despite the absence of any observable effect of the deletion of *BXI1* on the activity of BCL-2 proteins, we further tested whether their activity could be affected by the deletion of *MIR1*. This gene encodes for a major mitochondrial phosphate carrier isoform [33] and was suggested to be indispensable for the effect of Bxi1p (Ybh3p) on Bax activity [22]. Besides the minute effect on the growth rate of *∆mir1* cells on galactose-based media, we did not observe any effect of *MIR1* deletion on the growth of BCL-2 family proteins expressing cells, neither in the presence nor in the absence of *BXI1*. Thus, Mir1p appears not to play a role in the permeabilization of membranes by pro-apoptotic BCL-2 proteins, neither in the cell-rescuing activity of anti-apoptotic BCL-2 proteins and by extension also in any downstream pathway that affects the survival upon expression of BCL-2 family proteins.

It has been somehow contradictory suggested that Bxi1p is involved in the response to some forms of stress not related to the action of the BCL-2 family. These include the regulated cell death induced by the treatment of cells with hydrogen peroxide or acetic acid, in which the activity of Bxi1p (Ybh3p) was reported to promote cell death [22]. Similarly, the presence of Bxi1p was shown to be required for the toxicity of ectopically expressed human TDP-43, which is a protein found deposited as inclusions in the brain of patients with amyotrophic lateral sclerosis (ALS) [34]. On the other hand, Bxi1p supports survival under conditions of endoplasmic reticulum stress [21]. No effect of deletion of *BXI1* was observed in the case of regulated cell death induced by hexadecenal [35]. Cells with either deleted or overexpressed *BXI1* grow slightly worse in the media containing elevated concentrations of copper [36]. Contrary to the former [22], in our experiments, we did not observe the protective effect of deletion of *BXI1* against the treatment with hydrogen peroxide and acetic acid. Moreover, we observed the increase in sensitivity of *∆bxi1* strain to treatment by acetic acid. Thus, our results are consistent with the idea that Bxi1p is involved in the pathway that responds to some forms of stress to promote cell survival, most likely by acting as a Ca^2+^ channel in the membrane of the endoplasmic reticulum [37,38]. Although one has to be careful because the extrapolation from mammalian cells may be dubious, this may be also in agreement with the observation of the effects associated with changes in intracellular Ca^2+^ in BI-1 knock-out mouse [39]. The fact that Bxi1p may be a part of a complex signaling pathway, rather than act directly in some specific process, is also supported by our preliminary results, showing that the effects of deletion of *BXI1* on survival in different stress conditions, including the endoplasmic reticulum stress, is strongly strain specific (Polčic et al. unpublished data) and may therefore depend on the activity of many other proteins.

The inhibition of Bax in yeast by overexpression of BI-1 homologs from many organisms, including *BXI1,* has been well established [14,20]. The mechanism of this inhibition remains to be uncovered. Our data hardly support the model, in which the inhibition relies on the direct interaction of Bxi1p with Bax or other BCL-2 family proteins. This inhibition could thus result from the overactivation of the stress response pathway, e.g., by increased Ca^2+^-release from the endoplasmic reticulum, downstream of the permeabilization of membranes by Bax.

## 4. Materials and Methods

### 4.1. Strains, Plasmids, and Growth Conditions

The wild-type yeast strain used throughout the study is *Saccharomyces cerevisiae* CML282 (*MAT**a** ura3-1*, *ade2-1*, *leu2-3*, *112; his3-11*, *15, trp1-Δ2*, *can1-100*, *CMV_p_(tetR-SSN6)::LEU2*) (kindly provided by Enrique Herrero, Universitat de Lleida) [40]. The strain CML∆bxi1, in which the complete coding sequence of *BXI1* is replaced with the KanMX4 marker gene (CML282 *bxi1::kanMX4*), was kindly provided by Nicanor Austriaco (Providence College). The deletion was verified by PCR.

Cells were grown on synthetic complete (SC) media containing the indicated carbon sources and lacking the appropriate amino acids or nucleobases. Yeast cells were transformed by standard lithium acetate protocols [41].

To delete the complete coding sequence of the *MIR1* gene, a linear fragment of DNA (deletion cassette), consisting of the *URA3* marker gene flanked by 5′UTR and 3′UTR sequences of *MIR1,* was prepared by PCR using oligonucleotides 5′-CAAGAAGAAGAACTACAAAGATCAAAAAGTCTCATCTCACCGTACGCTGCAGGTCGAC-3′ and 5′-GAGGAGAGAATATATATGCATGTATCAATCAAGACCATTTATCGATGAATTCGAGCTCG-3′ as primers and plasmid pJET-URA3 as a template. This plasmid was prepared by ligation of 1 kbp fragment of *URA3,* prepared by PCR using oligonucleotides 5′-CGTACGCTGCAGGTCGACGAAGGAAGAACGAAGGAAGG-3′ and 5′-ATCGATGAATTCGAGCTCGGGTAATAACTGATATAATT-3′ as primers and plasmid p416-MET25 [42] as a template, with pJET1.2/blunt vector (ThermoFisher Scientific, Waltham, MA, USA). After a transformation of yeast cells with a deletion cassette, deletants (CML∆mir1, CML∆bxi1∆mir1) were selected by cultivation on SC plates without uracil and verified by PCR.

To express murine Bax or Bak containing N-terminal HA (haemagglutinin) tag, plasmids containing the gene placed behind galactose inducible Gal1/10 promoter—pRS303-GHBax and pRS303-GHBak—were integrated into the *HIS3* locus in the chromosome [9,43]. To modulate the expression of Bcl-XL and Bcl-2, cells were transformed with plasmids pCM252-HA-BCL-XL or pCM252-BCL2 [9,11], grown in glucose-based SC media without tryptophan, washed, and transferred to selective liquid or solid media containing the indicated concentration of doxycycline.

### 4.2. Viability Tests

To assess the activity of BCL-2 family proteins, the growth potential of individual strains expressing corresponding mammalian genes was followed by a drop test. Cells were grown overnight in media containing 2% glucose, diluted to OD_600_ = 0.5, and 10 μL aliquots of serial 5-fold dilutions were spotted on to test plates containing the indicated concentration of galactose to induce the expression of Bax/Bak and doxycycline to induce the expression of Bcl-XL/Bcl-2. Growth was assessed following incubation at 28 °C for 2–4 days. Experiments were repeated at least three times, representative results are shown.

To assess the survival of cells after treatment with acetic acid, a protocol of Ludovico et al. [44] was followed. Briefly, cells were grown in complete YPG medium (1% yeast extract, 2% bactopeptone, 3% glycerol) until exponential phase, harvested by centrifugation, diluted to cell density corresponding to OD_600_ = 1 in a same medium supplemented with 80 mM acetic acid and pH adjusted to 3 (and to control medium with the same pH but without acetic acid), and incubated at 28 °C for three hours. Samples of acetic acid-treated and control cells were then spread on the Petri dishes containing complete YPD (1% yeast extract, 2% bactopeptone, 2% glucose) media, incubated for 2–3 days and colonies counted.

The survival of H_2_O_2_ treatment was tested according to Madeo et al. [45]. Cells were grown in YPG until the culture reached the absorbance OD_600_ = 0.5–0.6; the cell suspension was then split into two flasks, and H_2_O_2_ to a final concentration of 0.6 mM was added to one; cells were incubated at 28 °C and after 3 h, samples were spread on the YPD plates. Colonies were counted after 2–3 days of cultivation at 28 °C.

Results of cell survival after acetic acid and H_2_O_2_ treatment were analyzed using one-way analysis of variance (ANOVA) followed by a Tukey post hoc test. Statistical significance is indicated in the corresponding figure.

## 5. Conclusions

Contradicting studies have described the role of gene *BXI1/YBH3* in yeast *Saccharomyces cerevisiae*, suggesting that it may support survival in stress conditions or promote cell death by acting as a yeast BH3-only protein. Our data show that the absence of this gene in the deletion mutant does not affect the activity of ectopically expressed Bax, Bak, Bcl-XL, and Bcl-2 in a way BH3-only proteins would, nor is it required for their activity. On the other hand, we observed increased sensitivity of this mutant to treatment with acetic acid, a regulated cell death inducer. Together these results support the model, in which Bxi1p promotes cell survival by participation in stress signaling.

## Figures and Tables

**Figure 1 ijms-24-12011-f001:**
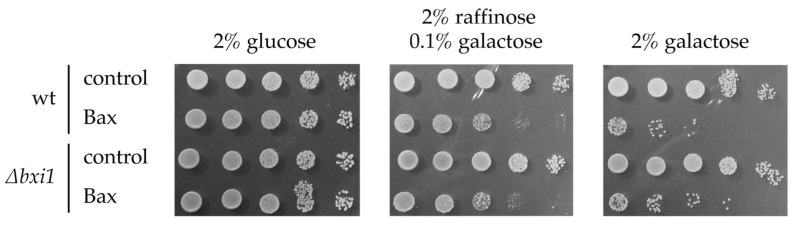
*BXI1* has no effect on the growth of wild-type and Bax-expressing yeast cells. CML282 (wt) and CML∆bxi1 (*∆bxi1*) cells not containing (control) or containing galactose inducible Bax gene (Bax) were cultivated in glucose media and cell suspensions were spotted onto SC plates containing indicated carbon source. The growth was assessed after 3 days.

**Figure 2 ijms-24-12011-f002:**
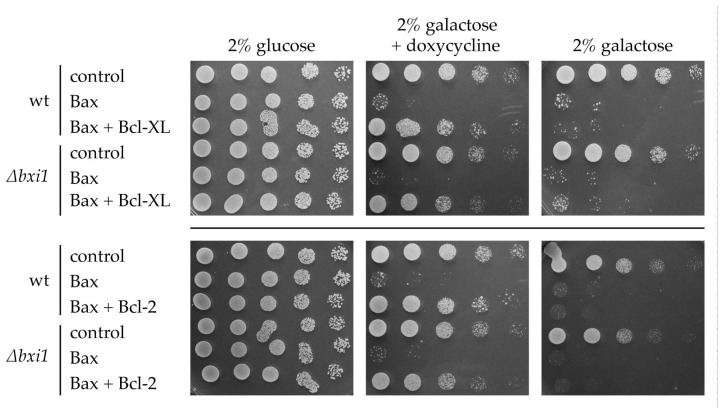
*BXI1* has no effect on the pro-survival activity of Bcl-2 and Bcl-XL. CML282 (wt) and CML∆bxi1 (*∆bxi1*) cells not containing (control) or containing galactose inducible Bax gene (Bax) were transformed with a plasmid containing doxycycline-inducible HA-Bcl-XL (Bcl-XL), Bcl-2 (Bcl-2), or with empty vector (not indicated). Cells were cultivated in glucose-containing SC media and cell suspensions were spotted onto SC plates containing indicated carbon source and concentration of doxycycline. The growth was assessed after 3 days.

**Figure 3 ijms-24-12011-f003:**
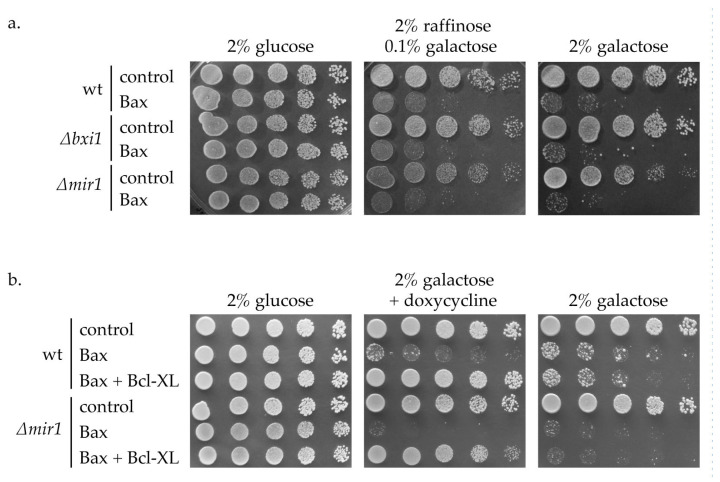
The absence of Mir1 does not affect the activity of proteins of the Bcl-2 family in yeast. (**a**) CML282 (wt), CML∆bxi1 (*∆bxi1*) and CML∆mir1 (*∆mir1*) cells not containing (control) or containing galactose inducible Bax gene (Bax) were cultivated in glucose media and cell suspensions were spotted onto SC plates containing indicated carbon source. (**b**) CML282 and CML∆mir1 cells not containing or containing galactose inducible Bax gene were transformed with plasmid containing doxycycline-inducible HABcl-XL (Bcl-XL), or with empty vector (not indicated). Cells were cultivated in glucose-containing SC media and cell suspensions were spotted onto SC plates containing indicated carbon source and concentration of doxycycline. The growth was assessed after 4 days.

**Figure 4 ijms-24-12011-f004:**
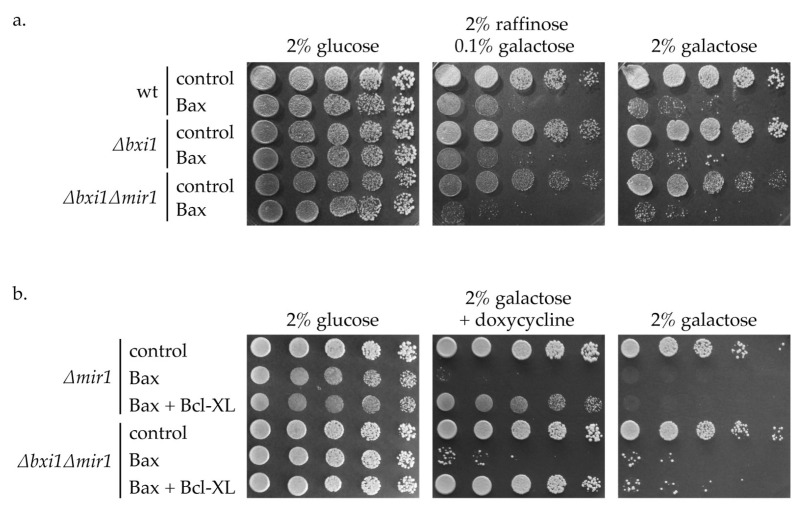
The absence of Mir1 does not affect the effect of Bxi1 on the activity of proteins of the Bcl-2 family in yeast. (**a**) CML282 (wt) CML∆bxi1 (*∆bxi1*) and CML∆bxi1∆mir1 (*∆bxi1∆mir1*) cells not containing (control) or containing galactose inducible Bax gene (Bax) were cultivated in glucose media and cell suspensions were spotted onto SC plates containing indicated carbon source. (**b**) CML282∆mir1 (*∆mir1*) and CML∆bxi1∆mir1 (*∆bxi1∆mir1*) cells not containing (control) or containing galactose inducible Bax gene were transformed with a plasmid containing doxycycline-inducible HABcl-XL (Bcl-XL), or with empty vector (not indicated). Cells were cultivated in glucose-containing SC media and cell suspensions were spotted onto SC plates containing indicated carbon source and concentration of doxycycline. The growth was assessed after 4 days.

**Figure 5 ijms-24-12011-f005:**
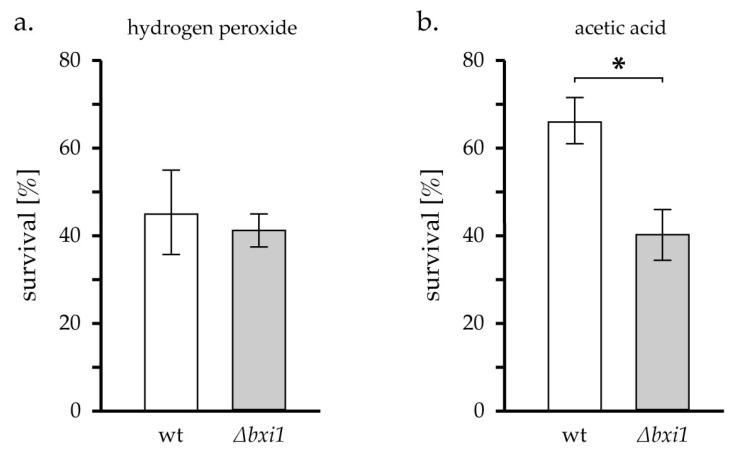
The survival of *∆bxi1* cells after treatment with acetic acid and hydrogen peroxide. Cells of the CML282 (wt) and CML∆bxi1 strain (*∆bxi1*) were treated with (**a**) 0.6 mM hydrogen peroxide or with (**b**) 80 mM acetic acid for 180 min, as described in the section Material and Methods. Survival corresponds to the ratio of the number of colonies formed by treated cells to the number of colonies formed by control (untreated) cells. The average of three independent experiments with standard errors is shown. Asterisk (*) indicates a statistical significance *p* < 0.05 of differences between mutants and the wild type.

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
