# Peer review of "Yeast Bax Inhibitor (Bxi1p/Ybh3p) Is Not Required for the Action of Bcl-2 Family Proteins on Cell Viability"

_ijms, 2023, doi:10.3390/ijms241512011_

Round 1
Reviewer 1 Report
The manuscript by Mentel et al. provides additionnal information about the Ybh3/Bxi1 protein, an homolog of the Bax inhibitors family of protein, that are also present in animals and plants. The manuscript clearly shows that Ybh3 is not required for Bax-induced yeast viability loss, nor for Bcl-xL-induced rescue. Furthermore, they observe no effect of the deletion of the mitochondrial phosphate carrier Mir1 on Bax and Bcl-xL effects. The experiments are rather straightforward and the conclusions are sound.
However, I think that the discussion should be more careful. The outcome of the experiments (growth/no growth) is "global" and gives no indication about subtle changes that might be induced by the absence of Ybh3 or Mir1. For instance, Bax and Bcl-xL mitochondrial localization, or Bax-induced release of cytochrome c might be changed, but not massively enough for having consequences of cell viability. Thus, sentences such as "our data not only indicate that BXI1 is not required for the activity of BCL-2 family proteins in yeast" (line 195) or "Mir1p thus appears not to play a role in the permeabilization of membranes by pro-apoptotic BCL-2 proteins" (line 205) should be toned down, and the authors should remind that cell viability does not provide a full view of the effects of Bax and Bcl-xL on mitochondria, and that more subtle changes may occur, that the authors might want to investigate in their future studies.
In the same spirit, I would change the title for something like "Yeast Bax Inhibitor (Bxi1p/Ybh3p) is not required for the action of Bcl-2 family proteins on cell viability" that would be more accurate.
Reviewer 2 Report
1-The abstract needs to be rewritten and the obtained results should be presented in general in the article.
2- The introduction needs to be modified and it is necessary to provide a summary of the reviewed articles.
3- The work method should be presented in a transparent manner.
4- The obtained results must be validated.
5- The obtained results should be discussed in comparison with others works.
6- The conclusion part is too weak. Add more in-depth discussion"
The manuscript need a through English proofreading.
Reviewer 3 Report
Regarding the manuscript entitled “Yeast Bax Inhibitor (Bxi1p/Ybh3p) is not required for the in vivo action of Bcl-2 family proteins”, the authors aimed to test the effect of the deletion of this gene on the pro-apoptotic activity of Bax and Bak and the anti-apoptotic activity of Bcl-XL and Bcl-2 as well on survival after treatment with inducers of regulated cell death in yeast, hydrogen peroxide and acetic acid.
The amount of the presented data is not enough to be accepted as an article. It can be accepted as a short communication after revision.
The study has no statistical analysis!! At least the results in Figure 5 can be statistically analyzed.
Please update your references list. Only one reference in your list was published in the last 3 years.
Round 2
Reviewer 3 Report
The manuscript can be accepted as a short communication.